



# Terrestrial runoff is an important source of biological INPs in Arctic marine systems

Corina Wieber[1,2], Lasse Z. Jensen[1,2,3], Leendert Vergeynst[2,4,5], Lorenz Maire[3,6,7], Thomas Juul-Pedersen[6], Kai Finster[1,2,8], Tina Šantl-Temkiv[1,2,3,8]

[1]Department of Biology, Microbiology, Aarhus University, Aarhus, 8000, Denmark
[2]iCLIMATE Aarhus University Interdisciplinary Centre for Climate Change, Roskilde, 4000, Denmark
[3]Arctic Research Centre, Aarhus University, Aarhus, 8000, Denmark
[4]Department of Biological and Chemical Engineering - Environmental Engineering, Aarhus University, Aarhus, Denmark
[5]WATEC, Centre for Water Technology, Aarhus University, Aarhus, 8200, Denmark
[6]Greenland Climate Research Centre, Greenland Institute of Natural Resources, Nuuk, 3900, Greenland
[7]Department of Estuarine and Delta Systems, Royal Netherlands Institute of Sea Research, Yerseke, 4401, The Netherlands
[8]Stellar Astrophysics Centre, Department of Physics and Astronomy, Aarhus University, Aarhus, 8000, Denmark

*Correspondence to*: Tina Šantl-Temkiv (temkiv@bio.au.dk)

**Abstract.** The accelerated warming of the Arctic manifests in sea ice loss and melting glaciers, significantly altering the dynamics of marine biota. This disruption in marine ecosystems can lead to the emission of biological ice nucleating particles (INPs) from the ocean into the atmosphere. Once airborne, these INPs induce cloud droplet freezing, thereby affecting cloud lifetime and radiative properties. Despite the potential atmospheric impacts of marine INPs, their properties and sources remain poorly understood. Analysing sea bulk water and the sea surface microlayer in two southwest Greenlandic fjords, collected between June and September 2018, and investigating the INPs along with the microbial communities, we could demonstrate a clear seasonal variation in the number of INPs and a notable input from terrestrial runoff. We found the highest INP concentration in June during the late stage of the phytoplankton bloom and active melting processes causing enhanced terrestrial runoff. These highly active INPs were smaller in size and less heat-sensitive than those found later in the summer and those previously identified in Arctic marine systems. A negative correlation between salinity and INP abundance suggests freshwater input as sources of INPs. Stable oxygen isotope analysis, along with the strong correlation between INPs and the presence of the bacterium *Aquaspirillum arcticum*, highlighted meteoric water as the primary origin of the freshwater influx, suggesting that the notably active INPs originate from terrestrial sources such as glacial and soil runoff.





## 1 Introduction

Climate change manifests in a steady rise in the global average temperature (IPCC, 2021a) and the Arctic region is particularly susceptible to its effects, experiencing a four times faster warming than the global average (Rantanen et al., 2022), a phenomenon known as Arctic amplification (Previdi et al., 2021). The consequences of the rise in temperature are severe, as

it leads to significant alterations in the energy balance (Letterly et al., 2018), changes that are manifesting in the loss of sea ice, glacial melt, and the thawing of permafrost (Previdi et al., 2021; Box et al., 2019; Chadburn et al., 2017). The reduction in sea ice exposes a larger fraction of seawater to the atmosphere, which facilitates the exchange of gases and aerosols between the ocean and the atmosphere.  The sea surface microlayer (SML), which is the interface between the ocean and the atmosphere with a thickness of less than a millimeter (Liss and Duce, 1997), plays a particularly important role in the ocean-atmosphere

exchange. In the SML, the concentration of organic material is often increased compared to the underlying sea bulk water (SBW) as surface-active compounds preferentially partition into the SML (Wurl et al., 2009). Previous studies have shown that the microbial community of the SML differs from the SBW just a few centimeters beneath the SML (Zäncker et al., 2018; Reunamo et al., 2011) and that specific bacterial taxa (e.g. *Flavobacteriaceae* and *Cryomorphaceae*) are enriched in the SML compared to the SBW (Zäncker et al., 2018). Atmospherically-relevant compounds comprising ice nucleating particles (INPs)

have also been found to concentrate in the SML compared to the SBW (Wilson et al., 2015; Hartmann et al., 2021).

INPs are particles that initiate the freezing of water at temperatures higher than the temperature of homogeneous freezing (-38 °C) and are of particular importance in the atmosphere where they impact ice-formation in clouds (Kanji et al., 2017). When aerosolized from the ground, INPs can namely be transported to high altitudes where they trigger the freezing of cloud

droplets and thus affect cloud radiative properties and lifetime. While mineral dust and soot are the numerically dominating atmospheric INPs at temperatures below -15 °C (Murray et al., 2012), biological INPs are dominating at temperatures above -15 °C and as high as -2 °C (Maki et al., 1974; Vali et al., 1976). Cloud ice formation has been frequently observed in Arctic mixed-phase clouds at temperatures, where only biological INPs are known to nucleate ice at atmospherically relevant concentrations (Griesche et al., 2021; Creamean et al., 2022). Therefore, there has recently been an increased focus on

quantifying biological INPs in both source environments and the atmosphere in the Arctic (Hartmann et al., 2021; Šantl-Temkiv et al., 2019; Creamean et al., 2022; Pereira Freitas et al., 2023). Especially during summer, when the long-range transport into the Arctic atmosphere is limited, locally sourced biological INPs may play an important role in cloud processes (Griesche et al., 2021). As cloud processes feed into the energy balance in the Arctic, and therefore modulate Arctic amplification (Serreze and Barry, 2011; Tan and Storelvmo, 2019), there is a need to improve our understanding of the sources,

controlling factors, and emission rates of cloud-relevant particles such as INPs.



Seawater, and in particular the SML, may act as a source of biological aerosols (bioaerosols) and INPs as wave breaking and bubble bursting can inject a significant amount of biological material and INPs into the atmosphere (Ickes et al., 2020; Wilson et al., 2015). So far it is unknown which types of INPs are responsible for the ice nucleation activity in seawater, but the sources may be linked to indigenous processes performed by marine microorganisms or to external inputs of terrestrial material into marine systems. Therefore, both marine and terrestrial ice nucleation active (INA) organisms may play a role when it comes to marine emissions of INPs. There is also an interplay between these two factors, as terrestrial runoff of nutrients has been shown to enhance the activity of marine microbes (Arrigo et al., 2017). Biological INPs can stem from a variety of sources such as different bacterial (Joly et al., 2013; Maki et al., 1974; Šantl-Temkiv et al., 2015), microalgal (Tesson and Šantl-Temkiv, 2018), and fungal species (Fröhlich-Nowoisky et al., 2015; Kunert et al., 2019), as well as lichen (Eufemio et al., 2023; Kieft and Ruscetti, 1990), viruses (Adams et al., 2021), pollen (Gute and Abbatt, 2020) and subpollen particles (Burkart et al., 2021), which inhabit marine and terrestrial environments. While we have substantial knowledge of bacterial ice nucleation proteins (Hartmann et al., 2022; Roeters et al., 2021; Garnham et al., 2011; Govindarajan and Lindow, 1988), our understanding of INA material excreted by other microorganisms, i.e. microalgae and fungi, is limited.

Recent studies have shown a correlation between biological INPs in Arctic seawater and the phytoplanktonic growth season (Creamean et al., 2019; Zeppenfeld et al., 2019). If the INPs originate from microorganisms associated with phytoplanktonic blooms their impact on atmospheric processes could become more pronounced with ongoing climate change as e.g. primary productivity is stimulated by higher temperatures and increased $CO_2$ levels. In addition, the melting of sea ice prolongs the phytoplankton growth season due to increased penetration of shortwave radiation into the water column (Park et al., 2015). This leads to more planktonic biomass and affects the marine ecosystem. Arrigo et al. (2008) observed an increase in annual primary production by marine algae of 35 Tg C $yr^{-1}$ between 2006 and 2007 due to decreasing sea ice coverage and a longer phytoplankton growth season. Consequently, the increased primary production induced by warming could impact the number of biogenic INPs released from the seawater and affect atmospheric processes. Further, terrestrial ice melt and runoff have been increasing over time (IPCC, 2021b). Studies have shown that glacial outwash sediment (Tobo et al., 2019; Xi et al., 2022), rivers (Knackstedt et al., 2018), thawing permafrost and thermokarst lakes (Creamean et al., 2020; Barry et al., 2023) contain high concentrations of INPs active at high sub-zero temperatures. As the runoff processes are enhanced, this will lead to increased inputs of highly active terrestrial INPs into the marine environments. Due to our poor understanding of biological INPs found in marine environments, it remains unclear whether it is indigenous microbial processes or external terrestrial inputs that dominate the pool of INPs in seawater.

Although prior studies have examined the concentrations of INPs in both bulk water and the microlayer, along with the influence of phytoplankton blooms on atmospheric INPs and INP concentrations in bulk water, the complete understanding of the dynamics and origins of marine INPs remains elusive. Hence, our research delved into the concentrations and properties of INPs within both the bulk water and the microlayer of southwest Greenland during the late and post-bloom seasons. We



chose to work on fjord systems to address the roles of indigenous processes versus terrestrial runoff as fjords are semi-closed systems where water circulation and therefore dilution of freshwater inputs is restricted. We correlated INP concentrations with chlorophyll a alongside the assessments of microbial composition, diversity, and abundance to understand whether INPs
are linked to the abundance of specific microbes, indicating their indigenous production. Finally, we correlated INP concentrations with salinity and $\delta O^{18}$ analysis to pinpoint the contribution of terrestrial inputs to the marine INP pool. This holistic approach aims to enhance our understanding of the dynamics and characteristics of biological marine INPs in the low Arctic.

## 2 Materials and methods

**2.1 Sample collection**

Sea surface microlayer and sea bulk water samples were collected at Kobbefjord (KF, N64°09.228, W51°25.906) and Godthåbsfjord (GF, N64°20.794, W51°42.709) in southwest Greenland (Fig. 1) in June, July, and September 2018. Approximately 100 mL of SML were collected per sample using a glass plate sampler (Harvey and Burzell, 1972). The glass plate sampler was immersed vertically and retracted at approximately 5 cm s⁻¹. The seawater was allowed to run off before the
adherent SML samples were scrapped of into a sterile bottle using a neoprene wiper.

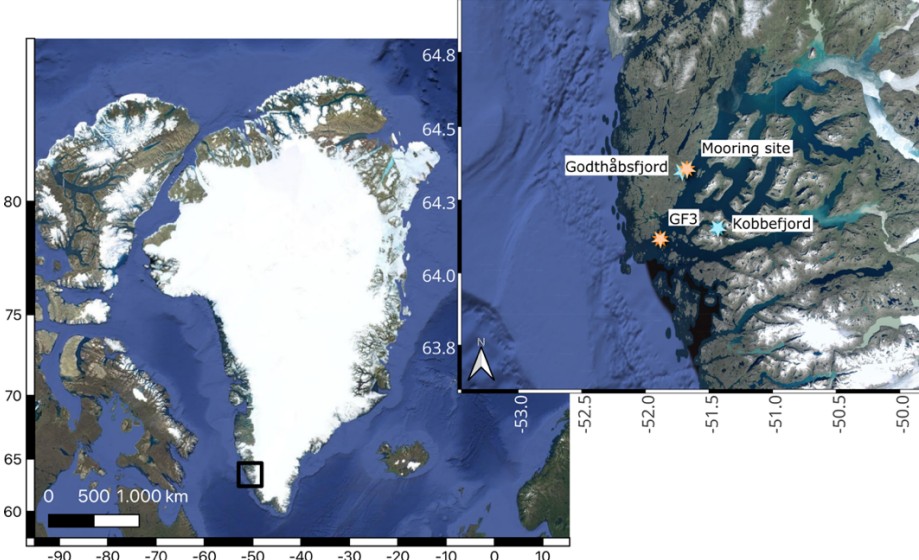

**Figure 1: Map of Greenland with a zoomed in view on the region around Nuuk. The sampling stations for SBW and SML samples Kobbefjord and Godthåbsfjord (blue) as well as the two stations for the chlorophyll and nutrient measurements GF3 and the**
**mooring station (orange) are indicated by stars. The map was produced in QGIS using the publicly available © Google Maps Satellite data layer.**



SBW samples were collected simultaneously by lowering a sterile bottle below the SML. Care was taken that the bottle was opened below the SML and closed before retrieving it back onto the boat. Samples for microbial community analysis were

immediately mixed 1:1 with a high-salt solution containing 25 mM sodium citrate, 10 mM EDTA, 450 g l$^{-1}$ ammonium sulphate, pH 5.2, (Lever et al., 2015) for the preservation of RNA. The samples were taken back to the lab immediately after sampling where they were concentrated onto Sterivex filters (0.22 µm), fixed in the presence of 1 ml of RNA later (Sigma Aldrich, US), and frozen at – 20 °C until the analysis.

**2.2 Measurement of ice nucleation activity**

Ice nucleation analysis was conducted using the micro-PINGIUN instrument (Wieber et al., 2024). Micro-PINGUIN is an instrument that allows droplet freezing assays using 384-well PCR plates. To ensure optimal thermal contact between the PCR plate and the cooling unit, a gallium bath is heated to 40 °C, and the PCR plate is immersed in the liquid gallium. The instrument is then cooled to 10 °C until the gallium solidifies, creating effective contact between the PCR plate and the surrounding

cooling unit. Once the plate is mounted, the samples are added to the wells to prevent sample heating during the melting process of the gallium, which could negatively impact the ice nucleating activity. For each sample, 80 wells of the 384-well PCR plates were filled with 30 µl of the sample each using an automatic 8-channel pipette (PIPETMAN P300, Gilson, US). The remaining 64 wells were filled with MilliQ water filtered through a 0.22 µm PES filter, serving as a negative control. The system is then cooled at a rate of 1 °C min$^{-1}$ until reaching -30 °C, while the freezing processes are recorded by an infrared

camera (FLIR A655sc/25° Lens; Teledyne FLIR, US).

The salt concentration for each sample was measured using a refractometer (WZ201, Frederiksen scientific, Denmark) and the freezing curves were corrected for the freezing point depression $\Delta T_{\mathrm{f}}$ using the theoretical formula for sodium chloride solutions:

$$\Delta T_{\mathrm{f}} = n\, c\, E_{\mathrm{f}} = n\, c \left(-1.86\, \tfrac{\mathrm{K*kg}}{\mathrm{mol}}\right),$$

$c$ is the molarity of the salt, $n$ is the number of ions of the dissociated salt (n = 2 for NaCl), and $E_f$ is the cryoscopic constant of water (Schwidetzky et al., 2021). The contribution of other components such as sulfate, magnesium, and calcium to the freezing point depression was considered minor and therefore neglected. 10-fold serial dilutions for the ice nucleation

measurements were prepared using autoclaved sodium chloride solutions closest to the salinity of the original seawater samples (5, 10, 15, 20, 25, 30, or 35 g kg$^{-1}$).



### 2.2.1 Heat treatment of the INP samples

Heat treatments were performed using a water bath. Following the initial ice nucleation test, PCR plates from the ice nucleation assay were covered with adhesive plastic foil to prevent cross-contamination of the samples during removal as well as evaporation of water during the heat treatment. The plates were then placed in a preheated water bath at 48 °C for 30 minutes. Subsequently, the plastic foil was removed, and ice nucleation activity was measured. The heat treatment was repeated the same way at 88 °C for 30 minutes and ice nucleation activity was remeasured a final time.


### 2.2.2 Filtration of the water samples

During the sampling campaign, filtrates for all samples were prepared with a 0.22 µm PES filter. Ice nucleation tests were conducted on all these samples. Selected samples (one of the duplicates) underwent a detailed investigation of ice nucleation
activity across different size ranges. Thus, prefiltered samples were loaded into Vivaspin filters (Vivaspin 20, Sartorius, Germany) with molecular weight cut-offs (MWCO) ranging from 1000 kDa to 100 kDa. Prior to use, Vivaspin filters were prewashed with 5 ml of MilliQ water to remove observed salt residues.

### 2.3 The fraction of meteoric water derived from stable oxygen isotopes


The fraction of stable oxygen isotopes $\delta^{18}O$ can be used to determine the contributions of sea ice meltwater (SIM) and meteoric water (MW), originating from precipitation e.g. rivers and glacial melt water, within a sample. This approach has previously been used and proven valuable to distinguish between SIM and MW in the Arctic Ocean (Burgers et al., 2017; Irish et al., 2019; Alkire et al., 2015; Yamamoto-Kawai et al., 2005). Following the approach used by Irish et al. (2019) and Burgers et al.
(2017) the water volume fractions of sea ice melt water ($f_{SIM}$), meteoric water ($f_{MW}$), and seawater ($f_{SW}$) were calculated using the following equations:

$$f_{SIM}S_{SIM} + f_{MW}S_{MW} + f_{SW}S_{SW} = S_{obs}$$
$$f_{SIM}\, \delta^{18}O_{SIM} + \, f_{MW}\, \delta^{18}O_{MW} + f_{SW}\, \delta^{18}O_{SW} = \delta^{18}O_{obs}$$
$$f_{SIM} + f_{MW} + f_{SW} = 1,$$

where $S$ is the corresponding salinity of the sample and $\delta^{18}O$ the ratio between $^{18}O$ and $^{16}O$ in water molecules. These equations assume that each sample consists of sea ice melt water, meteoric water, and a seawater reference and are derived from the



conservation equations. Net sea ice formation results in negative $f_{SIM}$ values (Alkire et al., 2015). For the sea ice melt water, we assume a salinity of 4 g kg⁻¹ and a $\delta^{18}O$ of 0.5 ‰ and for the meteoric water values of 0 g kg⁻¹ and -20 ‰, respectively (Irish et al., 2019; Burgers et al., 2017). Further, we assume that our samples are mainly influenced by the west Greenland current waters and thus we chose the values for the reference seawater as $S_{SW}$ = 33.5 g kg⁻¹ and $\delta^{18}O_{SW}$ = -1.27 ‰ (Burgers et al., 2017). $\delta^{18}O$ values were measured with PICARRO Li-1102 (18O). $\delta^{18}O$ values correspond to the deviation from the Vienna Standard Mean Ocean Water (V-SMOW) in permille (‰). Measurements were calibrated using three internal reference water samples (-55.5 ‰, -33.4 ‰, and -8.72 ‰) and the average value of 5 to 6 replicate injections was taken for the calculations. Standard deviations between replicate measurements are ranging between 0.004 ‰ to 0.12 ‰.

**2.4 Nutrient and chlorophyll a concentrations**

Nutrient concentrations were extracted from the database of the Greenland Ecosystem Monitoring (GEM) Project (https://data.g-e-m.dk). Measurements were carried out at the MarineBasis in Nuuk (location GF3 in Fig.1). Nitrate and nitrite concentrations were measured by vanadium chloride reduction (Greenland Ecosystem Monitoring, 2020a), phosphate (Greenland Ecosystem Monitoring, 2020b) and silicate (Greenland Ecosystem Monitoring, 2020c) concentrations were measured spectrophotometrically. Additionally, chlorophyll a measurements from 1 m depth were extracted from the GEM database. After collection, the seawater samples were filtered, and the filters were stored frozen in 10 ml 96% ethanol until analysis with a Turner TD-700 Fluorometer. The same method was applied for seawater sampled at 12 m depth in Kobbefjord during the sampling dates of this study. For chlorophyll measurements from the mooring site (Fig. 1) a fluorescence sensor (Cyclops-7 Logger, PME) was deployed at 5 m depth from March to October 2018. The sensor was calibrated with chlorophyll standards (Turner Design). To prevent biofouling, the instruments were wrapped with copper tape. Upon retrieval, data was read out and despiked using the OCE package (Kelley et al., 2022) and chlorophyll a concentrations were calculated as a 3-day average.

**2.5 DNA Extraction, Quantitative Polymerase Chain Reaction (qPCR) and amplicon sequencing**

As freezing of the Sterivex filters, containing the high-salt solution, lysed some microbial cells and thus released their nucleic acids (data not shown), we used two different protocols for DNA extraction, one for the DNA in solution and one for the DNA which was still within the intact cells on the filter. In brief, the high-salt solution was extracted from the Sterivex filter into a separate tube. Then, DNA was purified using the CleanAll RNA/DNA Clean-up and concentration Kit (Norgen Biotek) following the manufacturers protocol. A combination of chemical and physical lysis was used on the Sterivex filters for simultaneously extraction of DNA as described by Lever et al. (2015). Last, the purified DNA from the Norgen Purification was pooled with the corresponding DNA from the Lever et al. (2015) extraction. qPCR was performed to quantify the amount





of bacterial 16S rRNA gene copies (DNA) as described earlier (Jensen et al., 2022), while 18S rRNA gene copies were quantified using primers Euk345F (5'- AAGGAAGGCAGCAGGCG-3') and Euk499R (5'- CACCAGACTTGCCCTCYAAT-'3) (Zhu et al., 2005). 16S rRNA library preparation of DNA was performed as described in Jensen et al. (2022) while 18S rRNA library preparation was performed with slight modification: Primers TareukFWD1 (5'-

CCAGCASCYGCGGTAATTCC-3') and TAReukREV3 (5'- ACTTTCGTTCTTGATYRA-3') were used to amplify the V4 region of the small subunit (18S) ribosomal RNA (Stoeck et al., 2010). The PCR mixture contained 4 μl template DNA instead of 2 μl. The thermal cycling was run with touchdown PCR approach with an initial denaturation step at 95 °C for 3 minutes, 10 cycles with denaturation at 95 °C for 30 seconds, annealing at 57 °C for 30 seconds, elongation at 72 °C for 30 seconds, followed by 15 cycles with denaturation at 95 °C for 30 seconds, annealing at 47 °C for 30 seconds, elongation at 72 and a

final elongation at 72 °C for 5 minutes. PCR clean-up was performed as for the 16S rRNA PCR products. The second round of PCR was run for 10 cycles to incorporate overhang adapters and was run with the same conditions as the previous PCR with an annealing temperature of 57 °C. Products were cleaned, and the Nextera XT Index primers were incorporated in a third PCR reaction which was run for 8 cycles following the previous condition and an annealing temperature of 55 °C. The PCR products were quantified using a Quant-iT™ dsDNA BR assay kit on a FLUOstar Omega fluorometric microplate reader

(BMG LABTECH, Ortenberg, Germany), diluted and pooled together in equimolar ratios. The pool was quantified using the Quant-iT™ dsDNA BR assay kit on a Qubit fluorometer (Thermo Fisher Scientific, Waltham MA) and then sequenced on the Illumina MiSeq platform (Illumina, San Diego, CA) which produces two 300-bp long paired-end reads.

**2.6 Bioinformatic analysis**


Bioinformatic analyses were performed in RStudio 4.3.3. 16S and 18S sequence reads were processed following the same pipeline. Primer and adapter sequences were trimmed from the raw reads using cutadapt 0.0.1 (Martin, 2011). Forward and reverse read quality were plotted with the plotQualityProfile function from DADA2 1.21.0 (Callahan et al., 2016). Based on the read quality a trimming of 280 bp and 200 bp were set for the forward and reverse reads, respectively, using FilterAndTrim,

according to their quality. The fastq files were randomly subsampled to the lowest read number using the ShortRead package 1.48.0 (Morgan et al., 2009), resulting in 42375 reads per sample for the 16S and 63555 reads per sample for the 18S, respectively. The subsampling allows for a more accurate comparison of the richness of the different samples. Error models were built for the forward and reverse reads, followed by dereplication and clustering into ASVs (Callahan et al., 2017) with DADA2. The denoised forward and reverse reads were merged using the function mergePairs with default parameters with a

minimum overlap of 12 nucleotides, allowing zero mismatches. Sequence tables were made with the function makeSequenceTable. ASVs shorter than 401 and longer than 430 nucleotides were removed from the 16S whereas sequences shorter than 360 and longer than 400 nucleotides were removed for the 18S dataset followed by chimeric sequence removal using the removeBimeraDenovo function. Taxonomic assignment was accomplished using the naive Bayesian classifier



against the SILVA ribosomal RNA gene database v138 (Quast et al., 2012) for the 16S sequences, while the 18S sequences

were classified against the Protist Ribosomal Reference (PR2) database v5.0.1 (Guillou et al., 2013) with the assignTaxonomy function from DADA2, and species assignment was performed with the assignSpecies function from DADA2. ASVs mapped to mitochondria, chloroplasts and metazoa were removed from the dataset. Samples were decontaminated using the prevalence method (Threshold = 0.1) from the Decontam package (Davis et al., 2018). Statistical tests and visualization of the data was performed with phyloseq (Mcmurdie and Holmes, 2013), Vegan (Dixon, 2003) and microeco (Liu et al., 2021).

**3 Results and Discussion**

**3.1 Spring bloom and secondary bloom in summer 2018**

The chlorophyll a concentration, serving as a proxy for algal biomass (Creamean et al., 2019; Krawczyk et al., 2021; Huot et al., 2007; Hartmann et al., 2021), increased from early April to late May with a brief decline in the end of April (Fig. 2a). In late August, there was a subsequent increase in chlorophyll a concentration lasting for two to three weeks. These periods of

increased chlorophyll a concentration align closely with times of nutrient scarcity, particularly a reduction in silicate levels, indicating that the nutrients were likely consumed by phytoplankton, most probably diatoms (Fig. 2b). The abundance and composition of nutrients, specifically carbon (C), nitrogen (N), and phosphorus (P), determine the activity and composition of the phytoplankton community (Arrigo, 2005). Increased primary production and chlorophyll a concentrations were previously shown to correlate with reduced nutrient concentrations such as nitrates ($NO_3^- + NO_2^-$), silicate ($SiO_2$), and phosphate ($PO_4^{3-}$)

(Juul-Pedersen et al., 2015). Low levels of these nutrients can limit phytoplankton growth (Harrison and Li, 2007). Silicate is a constituent of diatom cell walls and thus a limiting factor for diatom cell growth (Bidle and Azam, 1999). Based on this data, we conclude the occurrence of a spring bloom (beginning of April to June) followed by a shorter secondary bloom at the end of August in the Nuuk region in 2018.



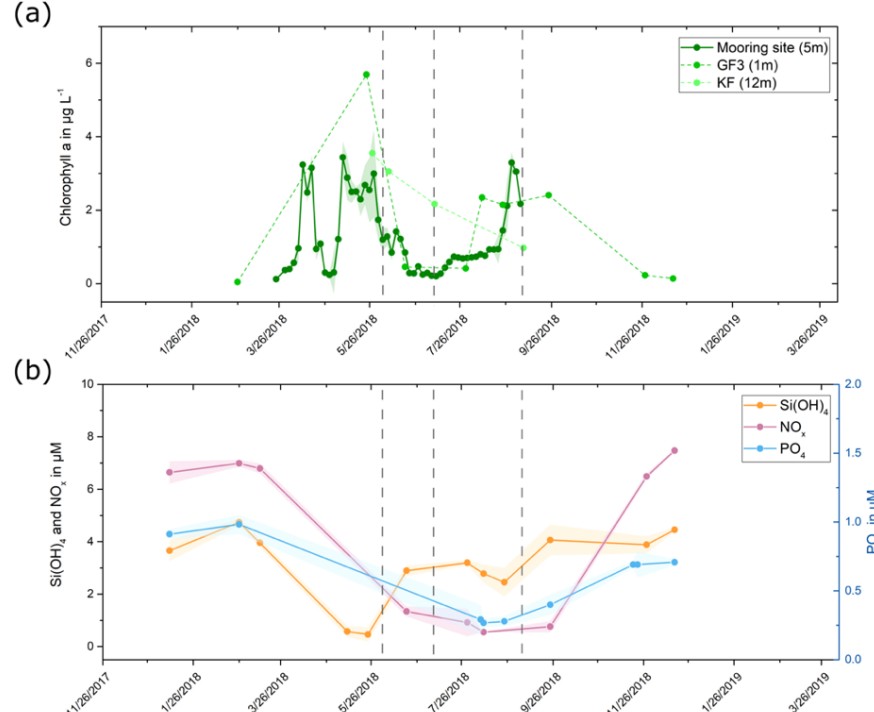

**Figure 2: (a) Chlorophyll a concentration in 2018. Data was collected at 3 measurement sites and data points for each site are connected by lines for visualization purposes. The green shaded area for the mooring site data shows the standard deviation for the 3-day average. (b) Silicate, nitrate and nitrite, and phosphate concentrations in 2018. Data points are connected by lines for visualization purposes and the shaded areas indicate the standard deviations for measurements at 1, 5, 10, and 15 meter depth. The sampling dates are highlighted by vertical dashed lines in both graphs.**

## 3.2 Seasonal variability and upconcentration of INPs in the SML

While freezing was initiated above -7 °C in all investigated SBW samples, the concentration of INPs active at -10 °C (INP$_{-10}$) cover a wide range from $1.3 \cdot 10^4$ INPs per liter to $6.1 \cdot 10^6$ INPs per liter. Typically, biological INPs are responsible for ice nucleation at temperatures higher than -15 °C (Murray et al., 2012), implying that the elevated onset freezing temperatures observed in our samples are attributable to INPs originating from biological sources. In addition, our study revealed enhanced INP$_{-10}$ concentrations in the SML compared to the corresponding SBW samples (Fig. 3b). This finding aligns with observations by Wilson et al. (2015) and Hartmann et al. (2021), whereas Irish et al. (2017) observed no significant upconcentration of INPs in the SML. This is unlikely to be related to the sampling technique as our study and the studies by Hartmann et al. (2021) and Wilson et al. (2015) observed an upconcentration of INPs in the SML despite utilizing different approaches for collecting the samples. Consequently, it is more plausible that these differences arise from spatial and temporal variations in the properties of the SML.





Given that the SML serves as the interface between water and air, specific characteristics of INPs, such as hydrophobic sites, may lead to their partitioning into the SML. Moreover, elevated concentrations of organic material in the SML (Wurl et al.,
2009) could provide energy for microorganisms in the SML and therefore lead to an increase in biogenic INP production, thereby explaining the higher INP concentrations observed in the SML.

Further, the ice nucleation tests revealed a seasonal variability with INPs having the highest onset freezing temperatures and significantly higher $INP_{-10}$ concentrations in June (Fig. S1). Notably, the INP concentrations in our study are generally higher
than those reported by Wilson et al. (2015), Hartmann et al. (2021), and Irish et al. (2017). While INP concentrations for July and September fall well within the range observed by Irish et al. (2019) and Creamean et al. (2019), SBW samples in June exceed the previously reported INP concentrations and show higher onset freezing temperatures. Previous studies (Wilson et al., 2015; Irish et al., 2017; Irish et al., 2019; Creamean et al., 2019) utilized 0.6-2.5 µl droplets for ice nucleation analysis, resulting in a 12-50 times higher detection limit (assuming the same number of investigated droplets) than our method.
Consequently, low-volume setups require higher concentrations of INPs for detection, thus not detecting highly active INPs that typically are present at lower concentrations. While small deviations in the reported freezing temperatures might occur due to methodological differences, INP concentrations in the two fjords observed in June are up to 4 orders of magnitude higher than previously reported values.

We found a moderate correlation (r = 0.60) between the concentration $INP_{-10}$ in the SBW and the in situ chlorophyll a concentration measured on the sampling dates in Kobbefjord. The correlation was stronger (r = 0.79) when focusing solely on the concentration of $INP_{-10}$ in Kobbefjord, where samples for quantifying chlorophyll and INP were collected at the same location and time (Fig. S2). Thus, INPs in June may originate from algal exudates released during the decay of phytoplankton in the late stage of the bloom. Factors such as nutrient limitation (Nagata, 2000), the transition between different growth stages
(Wetz and Wheeler, 2007), cell death, cell lysis, and excretion can enhance the release of dissolved organic matter (DOM) and dissolved organic carbon (DOC) by phytoplankton potentially including INA material (Thornton, 2014; Norrman et al., 1995). Aside from containing INA material (Ickes et al., 2020; Wilson et al., 2015), the released organics may also serve as nutrients for heterotrophic producers of INA material and may affect marine INP concentration by shaping the heterotrophic community composition and enhancing abundance and activity of INA microorganisms (Mühlenbruch et al., 2018). INPs produced by
aquatic microalgae are often reported to be active below -12 °C (Tesson and Šantl-Temkiv, 2018; Thornton et al., 2023), however, INPs from epiphytic bacteria and fungi can nucleate ice at higher temperatures (as high as -2 °C and -2.5 °C, respectively (Fröhlich-Nowoisky et al., 2015; Huffman et al., 2013; Pouleur et al., 1992a; Maki et al., 1974; Vali et al., 1976). As INP concentrations and onset freezing temperatures observed in June were higher than previously reported values, this might indicate that the highly-active INPs are originating from other, potentially terrestrial sources. The possibility of terrestrial
runoff serving as a source of biological INPs in seawater has also been previously explored (Irish et al., 2019).



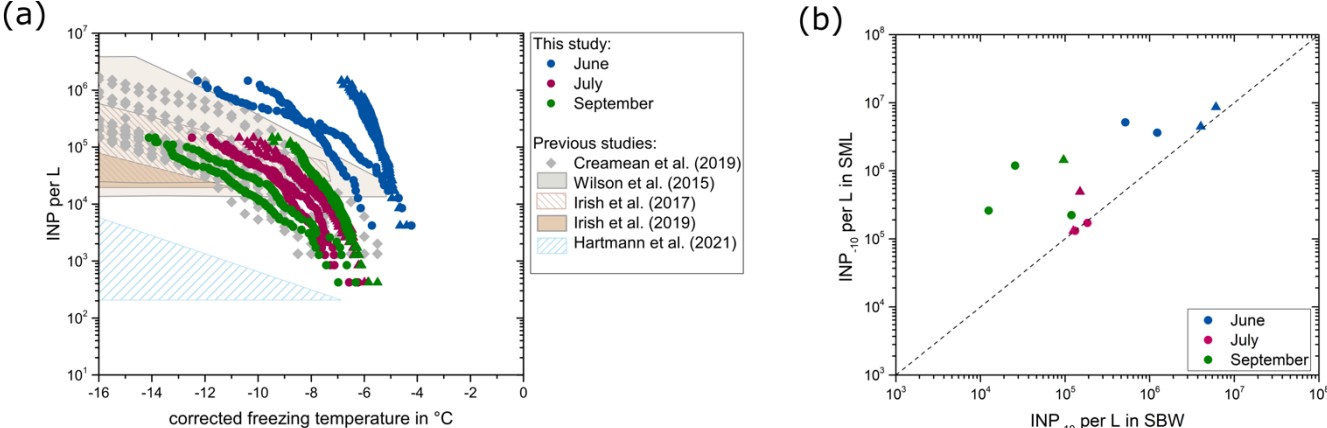

**Figure 3: (a) Number of INPs per L seawater for the bulk water samples collected in the Kobbefjord (circles) and Godthåbsfjord (triangles). The boxes represent the data ranges reported by previous studies and the grey data points represent the data reported**
**by Creamean et al. (2019). (b) Number of INPs active at -10 °C in the bulk water compared to the microlayer for Kobbefjord (circles) and Godthåbsfjord (triangles). The dashed line represents the 1:1 fraction.**

### 3.3 Shift in INP size and properties over time

To further characterize the INPs and to investigate their sources, we examined their size and heat sensitivity based on changes
in $T_{50}$ temperatures (temperature where 50% of the droplets were frozen). In June, $T_{50}$ temperatures remained constant until a molecular weight cut-off (MWCO) of 300 kDa, and decreased significantly after filtration with a 100 kDa MWCO (One-way ANOVA, $p < 0.05$, Supplementary Fig. S3). Only the SBW sample collected in June in KF showed a reduction in the freezing temperatures after filtration with an MWCO of 1000 kDa (Fig. 4). The data suggests that the prevailing size range of INPs in June falls between 100 kDa and 300 kDa. Contrary, in all samples collected in July, the $T_{50}$ temperatures decreased to below
-17 °C after filtration with an MWCO of 1000 kDa. Samples collected in September showed a size disparity between INPs from the SML and the SBW. SML samples showed higher freezing temperatures with INPs in the range of 300 kDa to 1000 kDa, while SBW samples followed the pattern observed in July, indicating INP sizes larger than 1000 kDa (Fig. 4). Overall, our data implies the presence of small biogenic particles (molecular weight < 300 kDa) in June, coinciding with the late stage of the phytoplanktonic spring bloom. In contrast, for samples collected after the spring bloom in July and September,
INPs are larger with a molecular weight greater than 1000 kDa and 300 kDa, respectively. Further, June samples showed no decrease in $T_{50}$ temperatures after moderate heating (48 °C), while samples from July and September were negatively affected by moderate heating (Fig. 5). Thus, the INPs with the ability to trigger freezing at high temperatures, primarily present in June, are smaller (100-300 kDa) and less heat sensitive (not affected by 48 °C).



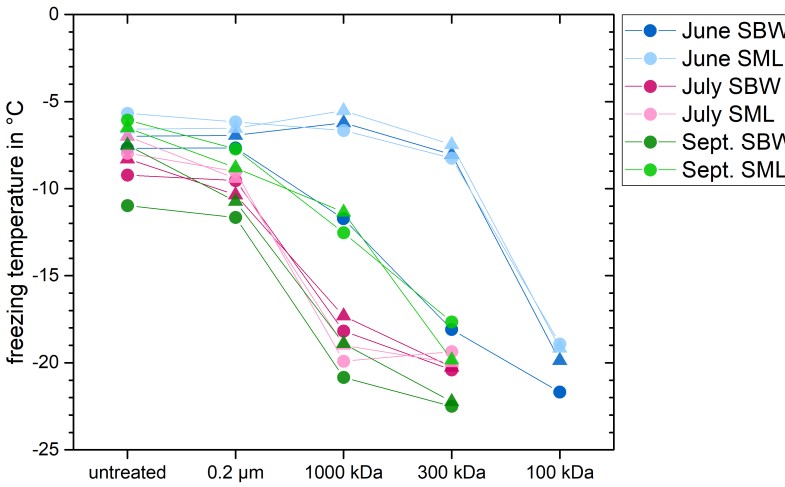


**Figure 4: Freezing temperatures for a fraction frozen of 0.5 ($T_{50}$ temperatures) after filtration with different pore size. Results from Kobbefjord are shown as circles and results from Godthåbsfjord are shown as triangles.**

The characteristics observed by heat treatments and filtration provide insights into the nature of the INPs. While Hartmann et
al. (2021) reported a removal of INPs with a 0.2 μm pore size filter in seawater collected from May to mid-July in the Arctic
Ocean close to Svalbard, Irish et al. (2017) and Wilson et al. (2015) found that marine INPs were not affected by filtration
through a 0.2 μm filter and could only be removed with a smaller pore size of 0.02 μm, approximately corresponding to a
300 kDa MWCO (Sartorius, 2022). Hartmann et al. (2021) suggest that their INPs were associated with larger particles, such
as bacteria, algae, fungi, or biological material attached to minerals. Irish et al. (2017) and Wilson et al. (2015) hypothesized
350 that the INPs they observed were small biological virus-size particles, probably phytoplankton or bacteria exudates, and that
they did not consist of whole cells or larger cell fragments. Similarly to Irish et al. (2017) and Wilson et al. (2015) we observed
small virus-size particles in July and August. During June, however, we observed a different type of INPs, which are smaller
than 300 kDa and have according to our knowledge not been previously reported in marine systems. Several studies have
shown that INPs that were washed off pollen grains and fungal cells are in the size range between 100-300 kDa (Pummer et
355 al., 2012; Pouleur et al., 1992b; Fröhlich-Nowoisky et al., 2015). Schwidetzky et al. (2023) showed that fungal INPs comprise
cell-free proteinaceous aggregates, with 265 kDa aggregates initiating nucleation at -6.8°C, while smaller aggregates nucleate
at lower temperatures. This points towards potential terrestrial sources of the INPs that we observed in June.





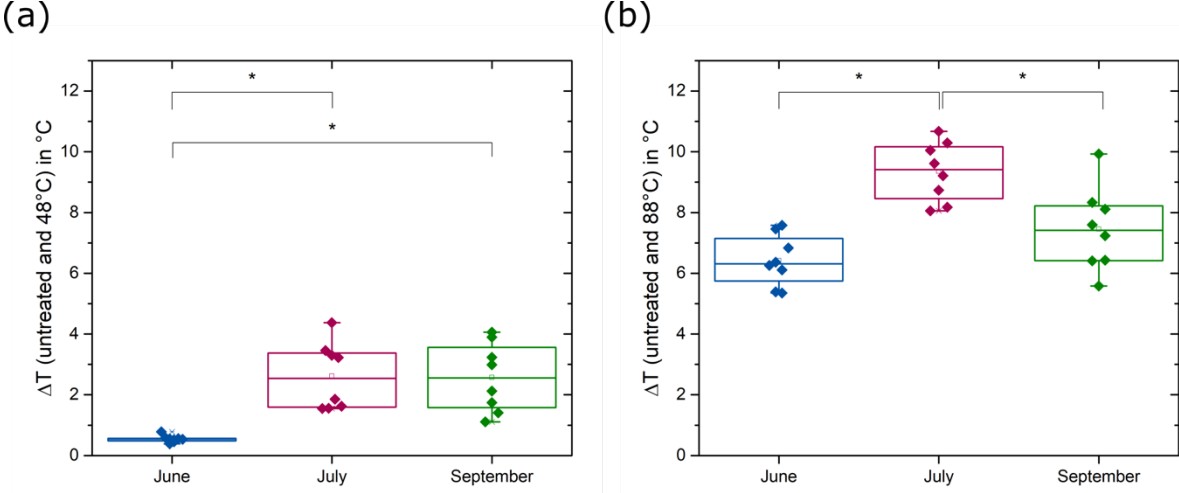

**Figure 5: Difference between the $T_{50}$ temperatures of untreated samples and samples heated to (a) 48 °C and (b) 88 °C. INPs in July and September are strongly influenced by moderate heating (48 °C), while INPs in June are less affected. All samples show significantly reduced $T_{50}$ temperatures after heating to 88 °C.**

Heat treatments are commonly performed to distinguish between biogenic INPs and inorganic INPs, assuming that the ice nucleation activity of biogenic INPs decreases when heated to sufficiently high temperatures, typically above 90 °C, due to protein denaturation (Daily et al., 2022). Although biological INPs can show distinct sensitivity to heating, heat treatments for environmental samples are often carried out at only one temperature (Daily et al., 2022, and the references therein). Few studies conducted extended heat treatments. Wilson et al. (2015) carried out heat treatments for SML samples for nine temperatures between 20 °C and 100 °C and observed decreasing activity with increasing heating temperatures. In combination with additional evidence, such as the results of filtrations, they attribute the ice nucleation activity to phytoplankton exudates present in the SML. D'souza et al. (2013) found significantly reduced freezing temperatures for filamentous diatoms from ice-covered lakes after heating to 45 °C. Hara et al. (2016) showed that the majority of ice nuclei from snow samples that were active above −10 °C, are inactivated at 40 °C, similar to those associated with *Pseudomonas syringae* cells. While bacterial INPs are proteinaceous and denature at temperatures above 40°C (Pouleur et al., 1992b; Hara et al., 2016), INPs from pollen, fungal spores, and lichen (Kieft and Ruscetti, 1990) are more heat-resistant. INPs in pollen washing water were thermally stable up to 112°C (Pummer et al., 2012) and INPs from fungal spores of *Fusarium sp.* and *Mortierella alpina,* as well as INPs from lichen, were proteinaceous and heat stable up to 60°C (Pouleur et al., 1992b; Fröhlich-Nowoisky et al., 2015; Kieft and Ruscetti, 1990). While the INPs that we found in July and September behaved similarly to INPs studied by Wilson et al. (2015), the INPs that we observed in June behaved similarly to what has previously been found for terrestrial INPs stemming from fungi, lichens, and pollen (Pouleur et al., 1992b; Fröhlich-Nowoisky et al., 2015; Kieft and Ruscetti, 1990).



We propose two alternative explanations for the seasonal variations in INP properties that we observed in the sampling period. Firstly, the difference in molecular weight could be due to different types of INPs present in June than later in summer. The observation that INPs in June are smaller and exhibit distinct responses to heat treatments compared to those observed later in the summer supports the idea that they represent distinct types of INPs. Based on their properties, INPs in June may have been released by pollen, fungal spores, or lichen in terrestrial environments and transported into the seawater by streams. INPs present in July and September may have alternative sources, e.g. indigenous microbial processes in the fjord systems. Secondly, the increase in INP molecular weight could be due to aggregation processes in the seawater. Organic matter may agglomerate over time or accumulate in transparent exopolymer particles (TEP), forming larger particles (Mari et al., 2017). TEPs are organic polymer gels primarily composed of heteropolysaccharides that form a hydrogel matrix (Engel et al., 2017). TEPs are highly adhesive and can enhance the aggregation of particles in water. Assembly and disaggregation processes result in different size ranges of the gels, typically exceeding 0.4 µm (Meng and Liu, 2016). Small INPs, which would typically pass through the filters used, might adhere to TEP and would be consequently removed by filtration.

**3.4 Correlation of INP concentration to environmental variables, microbial abundance, and community composition**

We performed qPCR and amplicon sequencing to link the abundance and diversity of bacteria and algae to the types and concentration of various INPs that we observed in the samples. Canonical Correspondence Analysis (CCA) was utilized to explore the correlations between environmental factors (salinity, chlorophyll a) and the microbial community compositions in both SBW and SML samples.

The eukaryotic community (Fig. S4 and Fig. S5), as derived from the 18S rRNA data, was dominated by Dinophyceae such as *Nusuttodinium* (high abundance in June) and *Gyrodinium* (high abundance in July). The bacterial community (Fig. S6 and Fig. S7) was dominated by the classes Bacteroidia and Gammaproteobacteria. On the genus level, a high abundance of *Polaribacter* was observed throughout all months. The presence of *Polaribacter* was found to correlate with the post-bloom and declining stage of the phytoplanktonic bloom in an Arctic fjord (Feltracco et al., 2021). For the 18S rRNA data, we found significantly increasing ribosomal gene copy numbers from June to September (Fig. S8). PerMANOVA analysis indicates a notable distinction among the months concerning the composition of the eukaryotic community ($p < 0.001$, Fig. S9). However, the observed alpha diversity was not significantly different (Fig. S10). The CCA analysis of the 18S rRNA data suggests that fewer organisms exhibit correlations with INP-10, and these correlations are weak (Fig. S11). The Mantel test assessing the correlation between dissimilarities in eukaryotic communities and environmental parameters demonstrated a significant correlation with salinity and chlorophyll a, whereas all other variables exhibited no correlation (Table S1).



The 16S rRNA analysis showed increasing 16S rRNA gene copy numbers from June to September with a significantly higher diversity of bacteria (observed and Shannon) in June (Fig. S12). A comparably high microbial diversity as we observed in June was previously reported for soils adjacent to Kobbefjord and may point to parts of the community introduced by terrestrial runoff (Šantl-Temkiv et al., 2018). Principal component analysis (PCA) followed by PerMANOVA demonstrated a significant difference between the bacterial community composition in June, July, and September ($p < 0.001$) (Fig. S13). These findings underscore the seasonal differences in the bacterial community structure with a lower copy number but higher diversity in June when INP concentrations are highest. Interestingly, although INPs were concentrated in the SML, bacterial cells did not show a similar pattern, as the differences in copy numbers and diversity between the SML and SBW were not significant.

The results of CCA for the 16S rRNA gene data are presented in Fig. 6. The bacterial communities in July and September exhibit a higher degree of similarity to each other than to the community observed in June. Additionally, the analysis demonstrated similarities between the communities in the two fjords. The Mantel test was subsequently conducted to assess the correlation between bacterial community dissimilarities (measured using robust Aitchison distance) and environmental parameters. Notably, the salinity exhibited strong correlations ($p = 0.003$), emphasizing its influential role in shaping the bacterial community composition (Table S2). Further, we found a strong negative correlation ($r = -0.81$, $p < 0.001$) between salinity and the concentration of $INP_{-10}$ with significantly lower salinity but higher concentration of INPs observed in June (Fig. 7a). These correlations suggest a strong impact of salinity within the observed fjords, pointing towards terrestrial runoff or melting sea ice as input of freshwater and potentially INPs.

We found a co-occurrence between three taxa, *Aquaspirillum arcticum*, *Colwellia sp.*, and *SUP05* (sulfur-oxidizing Proteobacteria cluster 05) and a high concentration of $INP_{-10}$ in the samples (Fig. 6). As confirmed by Pearson correlation, the co-occurrence was strongest between $INP_{-10}$ and the abundance of *Aquaspirillum arcticum* ($r = 0.90$, $p < 0.001$) followed by *Colwellia* sp. ($r = 0.83$ $p < 0.001$). *Aquaspirillum arcticum* is a psychrophilic bacterium with a low salt tolerance that was isolated (Butler et al., 1989) from snow and ice-covered Arctic sediment. Later, it is also found in low-saline Arctic environments, particularly in snow (Harding et al., 2011) and in melt pools (Brinkmeyer et al., 2004). Its strong correlation with $INP_{-10}$ concentrations in the fjords that we observe in our study implies that INPs are either produced by *Aquaspirillum arcticum* or that both INPs and *Aquaspirillum arcticum* cells were introduced to the fjords through terrestrial runoff from the same sources. *Colwellia sp.* is commonly found in sea ice and polar seas (Brinkmeyer et al., 2004). In previous studies it was demonstrated that *Colwellia sp.* strains isolated from sea ice have ice-binding properties due to the presence of antifreeze proteins (Hanada et al., 2014; Raymond et al., 2007). Structural similarity between antifreeze proteins and ice nucleating proteins was shown, although they have opposite functions (Davies, 2014). While the strong correlation between the relative abundance of *Colwellia* sp. and $INP_{-10}$ indicates that *Colwellia sp.* may produce ice nucleating compounds, this has not been validated yet. While both melting sea ice and terrestrial runoff could account for the input of freshwater and thus low salinities in June, a stronger correlation with the terrestrial *Aquaspirillum arcticum* points at terrestrial runoff as the key source.



The terrestrial run-off could either contain INPs produced by terrestrial microorganisms or it might provide nutrients to marine microorganisms, thereby enhancing microbial production of ice nucleation active material in the fjords (Irish et al., 2019; Irish et al., 2017; Meire et al., 2017; Arrigo et al., 2017). To further decipher whether terrestrial runoff or sea ice melt water was

driving the freshening of the seawater in the fjord, which correlated to high INP concentrations, we included analysis of the stable oxygen isotopes $\delta^{18}$O.

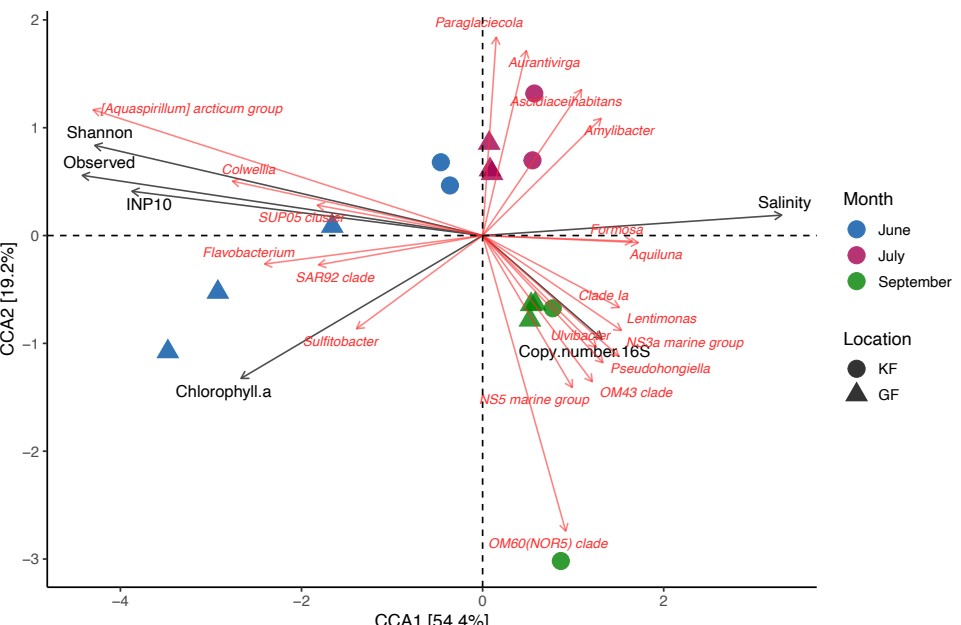

**Figure 6: Canonical correspondence analysis for the 16S rRNA data (20 taxa). Small angles between the arrows indicate a good correlation between the taxa and the external parameter while the length of the arrow is indicated of the importance of this parameter. As we observed a negative correlation between the salinity and the ice nucleation activity, these arrows point into opposite directions.**






## 3.5 Freshwater fractions from sea ice melt water and meteoric water


The freshwater fractions of sea ice melt water and meteoric water were calculated and correlated to the number of INPs active at -10 °C. A stronger correlation was found between the number of INP-10 and the fraction of meteoric water ($r = 0.84$, $p < 0.001$) as shown in Fig. 7b, while correlations with sea ice meltwater were weaker ($r = 0.64$, $p < 0.001$, Fig. S14), implying

a predominant influence of fresh water from melting glaciers and/or from precipitation.

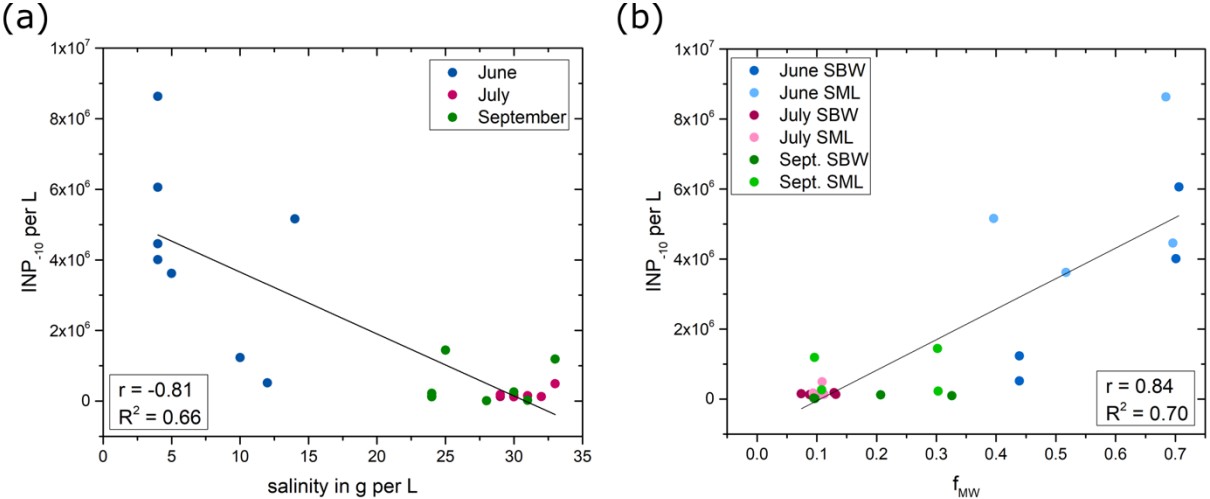

**Figure 7: The INP-10 concentration as a function of (a) salinity and (b) freshwater fractions from meteoric water. The lines represent
linear regressions of all data points shown in the graphs. Both correlations are significant (p < 0.001).**

The study region is surrounded by six glaciers, including three marine-terminating glaciers, supplying melt water and runoff into the fjords (Van As et al., 2014). A modeling study showed that melt and runoff in the Nuuk region doubled during the past two decades (Van As et al., 2014). The summer melt season in 2018 recorded exceptionally high surface melting across

the Greenland Ice Sheet, surpassing previous records in early June, late July, and early August (Osborne, 2018). Previous studies have shown that meteoric water, including glacial outwash sediment (Tobo et al., 2019; Xi et al., 2022), rivers (Knackstedt et al., 2018), thawing permafrost and thermokarst lakes (Creamean et al., 2020; Barry et al., 2023) contains high concentrations of INPs active at high sub-zero temperatures. Glacial outwash from the Arctic region was found to contain highly active organic INPs (Tobo et al., 2019). Xi et al. (2022) also demonstrated the presence of INPs active above -15°C in

glacial outwash sediments, albeit in smaller concentrations. Analysis of river samples revealed significant ice nucleation activity attributed to submicron-sized biogenic INPs, exhibiting comparable ice nucleus spectra to those produced by the soil fungus *Mortierella alpina*. These findings imply a terrestrial source for these INPs (Knackstedt et al., 2018). This supports the





conclusion that terrestrial runoff is a major source of marine INPs in the investigated region. With ongoing warming in the Arctic, microbial activity, and production of INP in terrestrial environments might increase, and thawing permafrost, glaciers,
and ice sheets may become of increasing importance as contributors of INPs to marine areas. Especially in fjord systems where the mixing with the open water is less pronounced terrestrial input might lead to increased INP concentrations in seawater and the SML. When these INPs get aerosolized from marine areas, they can trigger ice formation in clouds and in turn, impact the properties of clouds and thus their radiative forcing (Serreze and Barry, 2011; Tan and Storelvmo, 2019).

## 4 Conclusion

In this study, we investigated the ice nucleating particles in sea bulk water and in the sea surface microlayer samples in relation to phytoplanktonic blooms and terrestrial runoff in two fjords in southwest Greenland. We observed a high concentration of INPs in June which decreased in July and September. Filtration and heat treatments revealed a novel type of marine INPs in June, characterized by smaller sizes and lower heat sensitivity compared to INPs observed later in the summer and those previously identified in Arctic marine systems. Abundant INPs in June co-occurred with a low abundance of bacterial cells
characterized by a high taxonomic diversity characteristic of terrestrial ecosystems. We noted a robust inverse relationship between salinity and the abundance of INPs, indicating that freshwater inputs likely contribute to increased INP concentrations. Stable oxygen isotopes in the freshwater fractions point towards meteoric water as the major source of the freshwater, that could wash INPs and nutrients of terrestrial origin into the fjords. This was supported by the fact that INPs also strongly correlated with the presence of a terrestrial bacterium, e.g. *Aquaspirillum arcticum*. The timely co-occurrence with the
phytoplanktonic bloom is rather a correlation and not a causation of the INPs as the freshwater also contains nutrients that could stimulate the phytoplanktonic bloom. Vertical mixing of the water column may have diluted INP concentrations in the upper marine layer, resulting in decreased INP concentrations in the subsequent months. However, the types of INP observed in July and September were distinct from those in June, indicating that these came from another, potentially indigenous source. Based on several lines of evidence including the INP properties, the negative correlation with salinity, the stable oxygen isotope
analysis, the correlation with microbial diversity, and the co-occurrence of INP with terrestrial bacterial species, we conclude that the highly active and abundant INPs that we observed in seawater in June originate from a terrestrial source, such as glacial and soil runoff. The quantitative significance of terrestrial INPs in marine environments outside fjord systems and coastal areas, as well as the extent to which sea spray contributes to their total atmospheric fluxes, needs to be determined through further investigation.






**Appendix A1**

**Table A1: Overview of SBW and SML samples investigated in this study. All samples are collected and examined in duplicates.**

| location | date | type | $T_{50}$ (°C) | $INP_{10}$ ($L^{-1}$) | heat sensitive at 48°C | salinity (g kg$^{-1}$) |
|---|---|---|---|---|---|---|
| KF | 04/06/2018 | SBW1 | -7.68 | $5.16 \cdot 10^5$ | no | 12 |
| KF | 04/06/2018 | SBW2 | -7.28 | $1.23 \cdot 10^6$ | no | 10 |
| KF | 04/06/2018 | SML1 | -5.67 | $5.16 \cdot 10^6$ | no | 14 |
| KF | 04/06/2018 | SML2 | -5.44 | $3.62 \cdot 10^6$ | no | 5 |
| KF | 09/07/2018 | SBW1 | -9.22 | $1.31 \cdot 10^5$ | yes | 29 |
| KF | 09/07/2018 | SBW2 | -8.98 | $1.84 \cdot 10^5$ | yes | 29 |
| KF | 09/07/2018 | SML1 | -7.96 | $1.31 \cdot 10^5$ | yes | 30 |
| KF | 09/07/2018 | SML2 | -7.81 | $1.71 \cdot 10^5$ | yes | 30 |
| KF | 07/09/2018 | SBW1 | -10.97 | $1.25 \cdot 10^4$ | yes | 28 |
| KF | 07/09/2018 | SBW2 | -9.95 | $2.57 \cdot 10^4$ | yes | 31 |
| KF | 07/09/2018 | SML1 | -6.06 | $2.60 \cdot 10^5$ | yes | 30 |
| KF | 07/09/2018 | SML2 | -5.02 | $1.19 \cdot 10^6$ | yes | 33 |
| GF | 05/06/2018 | SBW1 | -6.99 | $4.01 \cdot 10^6$ | no | 4 |
| GF | 05/06/2018 | SBW2 | -6.90 | $6.06 \cdot 10^6$ | no | 4 |
| GF | 05/06/2018 | SML1 | -6.58 | $4.46 \cdot 10^6$ | no | 4 |
| GF | 05/06/2018 | SML2 | -5.90 | $8.63 \cdot 10^6$ | no | 4 |
| GF | 13/07/2018 | SBW1 | -8.30 | $1.25 \cdot 10^5$ | yes | 30 |
| GF | 13/07/2018 | SBW2 | -7.99 | $1.50 \cdot 10^5$ | yes | 31 |
| GF | 13/07/2018 | SML1 | -6.99 | $1.31 \cdot 10^5$ | yes | 32 |
| GF | 13/07/2018 | SML2 | -5.60 | $4.93 \cdot 10^5$ | yes | 33 |
| GF | 07/09/2018 | SBW1 | -7.50 | $9.59 \cdot 10^4$ | yes | 24 |
| GF | 07/09/2018 | SBW2 | -7.57 | $1.19 \cdot 10^5$ | yes | 24 |
| GF | 07/09/2018 | SML1 | -6.50 | $1.44 \cdot 10^6$ | yes | 25 |
| GF | 07/09/2018 | SML2 | -9.75 | $2.23 \cdot 10^5$ | yes | 24 |

*Data availability.* The data presented in this study are deposited in the European Nucleotide Archive under the accession number PRJNA1108919.

*Author contributions.* TST and KF designed and supervised the research project. LV collected the SML and SBW samples in Greenland. LM and TJP provided data for the chlorophyll concentrations. LZJ performed the microbial and bioinformatic 530 analysis. CW conducted the ice nucleation measurements and analysis and wrote the manuscript with contributions from the co-authors.



*Competing interests.* The authors declare that they have no conflict of interest.

*Acknowledgements.* We are grateful to Ana Sofia Ferreira for analyzing the chlorophyll satellite data. The authors thank Egon Randa Frandsen and Inge Buss la Cour for the measurements of the stable oxygen isotopes. Further, we would like to gratefully acknowledge Mette L.G. Nikolajsen and Britta Poulsen for their assistance in the laboratory. Data from the Greenland Ecosystem Monitoring Programme were provided by the Greenland Institute of Natural Resources, Nuuk, Greenland in collaboration with Department of Bioscience, Aarhus University, Denmark and University of Copenhagen, Denmark.


*Financial support.* This work was supported by The Villum Foundation (23175 and 37435), the Independent Research Fund Denmark (9145-00001B), The Novo Nordisk Foundation (NNF19OC0056963), The Danish National Research Foundation (DNRF106, to the Stellar Astrophysics Centre, Aarhus University), and The Carlsberg Foundation (CF21-0630).

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
