# Peer review of "Terrestrial runoff is an important source of biological INPs in Arctic marine systems"

_EGUsphere, 2024_

## Author Comment (AC1)

Referee #1

Comments:

1. Figure 1: Perhaps add photographs of the sites, so the reader gets an impression of the surroundings in which the samples were collected.

Thank you very much for this suggestion. We agree that it would be very nice to show photographs of the different sampling sites, unfortunately, no photographs were taken during the sampling campaign, so we have decided to not include photographs.

2. Section 3.2 discusses differences between SML and SBW in terms of INPs. Lines 276-279 state: "In addition, our study revealed enhanced INP-10 concentrations in the SML compared to the corresponding SBW samples (Fig. 3b). This finding aligns with observations by Wilson et al. (2015) and Hartmann et al. (2021), whereas Irish et al. (2017) observed no significant upconcentration of INPs in the SML." Yet, a closer look at Figure 3b reveals that one third of the samples does not show an upconcentration. So, better qualify the cited statement.

We have modified Fig. 3b showing the $T_{50}$ temperatures instead of the $INP_{10}$ values. In this representation the enhanced concentration of INPs in the SML becomes clearer. Further, we have added the fraction frozen curves for the undiluted samples of both fjords as well as a box plot showing a significant difference between the $T_{50}$ temperatures in the SML and the SBW to the Supplementary.

Changes in the text:

Line 282: "While freezing was initiated above -7 °C in all investigated SBW samples (Fig. S1), the concentration of INPs active at -10 °C ($INP_{-10}$) covered a wide range from $1.3 \cdot 10^4$ INPs per L to $6.1 \times 10^6$ INPs per L (Fig. 3a). Typically, biological INPs are responsible for ice nucleation at temperatures higher than -15 °C (Murray et al., 2012), implying that the elevated onset freezing temperatures observed in our samples are attributable to INPs originating from biological sources. In addition, our study revealed higher $T_{50}$ temperatures (temperature where 50% of the droplets were frozen) in the SML compared to the corresponding SBW samples (Fig. 3b, Fig. S2) showing that the highly active INPs are primarily found in the SML, which may affect their emissions into the atmosphere through wave breaking and bubble bursting (Ickes et al., 2020; Wilson et al., 2015)."

**Corrected Fig. 3:**

[Figure]

Figure 3: (a) Number of INPs per L seawater for the bulk water samples collected in the Kobbefjord (circles) and Godthåbsfjord (triangles). The INP data in June is derived from a 10-fold dilution due to the high activity. The boxes represent the data ranges reported by previous studies and the grey data points represent the data reported by Creamean et al. (2019). (b) Comparison of the T50 temperatures in the SBW in relation to the SML for Kobbefjord (circles) and Godthåbsfjord (triangles). The dashed line represents the 1:1 fraction.

**Figures added to Supplementary:**

[Figure]

Figure S1: Fraction frozen curves for the undiluted samples for the Kobbefjord (a) and the Godthåbsford (b). All samples are analyzed in duplicates.

[Figure]

Figure S2: T50 temperatures for the SBW samples in comparison to the SML samples shown as box plot with 25th and 75th percentiles. T50 temperatures are significantly higher in the SML ($p < 0.05$).

3. Can you extrapolate the trendlines in Figure 7b to get a rough estimate of what INP concentration might be in pure meteoric water (fMW = 1.0)? This number would allow for a more quantitative comparisons with INP concentration in other freshwaters discussed in lines 480-487

Thank you for this comment. It is an interesting idea to extrapolate the fraction of meteoric water. We have modified Fig. 7b accordingly and added the concentration of INPs for fMW=1.0 in Line 537-542:

"The fact that the study region is impacted by terrestrial runoff and glacial melt water as well as the strong correlation between the fraction of meteoric water and the $INP_{-10}$ concentration, supports the conclusion that terrestrial runoff is a major source of marine INPs in the investigated region. An extrapolation of the trendline in Fig. 7b leads to an estimated concentration of $7.8 \cdot 10^6$ INPs per L active at -10 °C in pure meteoric water ($f_{MW}$=1) which is in good agreement with the average concentration of $1.0 \cdot 10^7$ INPs per L active at -10 °C reported for freshwater samples from streams in eastern Greenland by Jensen et al. (2024)."

Jensen, L. Z., Simonsen, J. K., Pastor, A., Pearce, C., Nørnberg, P., Lund-Hansen, L. C., Finster, K., and Šantl-Temkiv, T.: Linking Biogenic High-Temperature Ice Nucleating Particles in Arctic soils and Streams to Their Microbial Producers, Aerosol Research Discuss., 2024, 1-29, 10.5194/ar-2024-18, 2024.

[Figure]

Figure 7: The $INP_{-10}$ concentration as a function of (a) salinity and (b) freshwater fractions from meteoric water. The lines represent linear regressions of all data points shown in the graphs. Both correlations are significant (p < 0.001).

---

## Author Comment (AC2)

Referee #2

Comments:

1. The authors propose two alternate explanations for the seasonal variations in INP properties proposed by the authors (Lines 382-393). The first explanation is that INPs in June may have been released by pollen, fungal spores, or lichen in terrestrial environments and transported into the seawater by streams. On the other hand, based on the analysis of the eukaryotic community derived from 18S rRNA data (Figs. S4, S5, and S8-S11, and Table S1), the authors suggest that fewer organisms exhibit correlations with INPs active at -10 degree C, and these correlations are weak (Lines 407-408). I think that there is a discrepancy between the first explanation and the 18S rRNA data. Why did the authors lead to the above first explanation?

Thank you very much for this comment. As previous studies have reported a link between the chlorophyll concentration/ abundance of microalgae and the INP concentrations we have chosen primers for the 18S rRNA analysis that target primarily marine microalgae. Thus, the fact that we do not see a correlation between the community derived from 18S rRNA data and the IN concentrations supports our conclusion that it is not the marine microalgae producing the INPs. Additionally, based on the literature available on INPs produced by pollen, fungal spores, or lichen we would expect a spatial decoupling between the INPs and the eukaryotes as the INPs are mainly released/excreted and therefore not necessarily found in the same environment. To clarify these findings, we have made the following changes:

Line 103: "We correlated INP concentrations with chlorophyll a alongside the assessments of microbial composition, diversity, and abundance to understand whether INPs are linked to the abundance of specific microbes, indicating either their indigenous production or acting as tracers for their source environments.".

Line 220: "Primers TareukFWD1 (5'- CCAGCASCYGCGGTAATTCC-3') and TAReukREV3 (5'- ACTTTCGTTCTTGATYRA-3') were used to amplify the V4 region of the small subunit (18S) ribosomal RNA targeting primarily the marine microalgae (Stoeck et al., 2010)."

Line 321: "Alternatively, the observed correlation between the chlorophyll a concentration and the $INP_{-10}$ concentration may not imply causality but may be attributed to the fact that terrestrial runoff simultaneously introduced INPs and nutrients, thereby enhancing the primary production in the fjords (Arrigo et al., 2017; Juul-Pedersen et al., 2015; Terhaar et al., 2021). In line with this, external inputs such as nutrients from terrestrial runoff can change the community composition (Ardyna and Arrigo, 2020) and stimulate primary production (Juranek, 2022)."

Further, we have made substantial changes to the section 3.4 "Correlation of INP concentration to environmental variables, microbial abundance, and community composition", to clarify the above mentioned comment as well as the comments 2 and 3. Please find the revised version of section 3.4 below:

"We performed qPCR and amplicon sequencing to link the abundance and diversity of bacteria and microalgae to the types and concentration of various INPs that we observed in the samples. While the 18S rRNA data aimed to decipher whether the INPs are linked to specific marine microalgae and thus likely produced indigenously in the seawater, the 16S rRNA data is used to both identify potential bacterial producers of INPs associated with phytoplankton blooms and to provide insights into the source environment of INPs transported from terrestrial environments. Canonical Correspondence Analysis (CCA) was utilized to determine correlations between environmental factors (salinity, chlorophyll a), the microbial community compositions, and the INP concentrations in both SBW and SML samples.

Considering the correlation we observed between the chlorophyll a and the $INP_{-10}$ concentration, we employed a community analysis of microalgae to search for potential indigenous producers of INPs. The microalgal community, as derived from the 18S rRNA data, is presented in Fig. S6 and Fig. S7. We identified several typical bloom-forming taxa such as centric diatom *Chaetoceros* (Biswas, 2022; Booth et al., 2002; Balzano et al., 2017), dinoflagellate *Gyrodinium* (Johnsen and Sakshaug, 1993; Hegseth and Sundfjord, 2008), and green algae *Micromonas* (Vader et al., 2015; Marquardt et al., 2016). We found a slight insignificant (p = 0.123) increase in the 18S rRNA gene copy numbers from June to September (Fig. S8). While the observed alpha diversity was not significantly different between the different months (Fig. S9), there was a significant distinction in the composition of the microalgal community (PerMANOVA, p < 0.001, Fig. S10). The CCA analysis shows a correlation of the microalgal community composition with salinity and chlorophyll a (Fig. S11, Table S1), implying that a combination of bloom dynamics and terrestrial runoff may have affected the communities. There was no correlation between the community composition and the $INP_{-10}$ concentrations (Table S1). We also investigated the association between specific microalgal taxa and the concentration of $INP_{-10}$ using Spearman's rank correlation analysis. While 22 taxa significantly correlated with the $INP_{-10}$ concentration (see Dataset (Wieber, 2024)) the majority of these taxa was present at low relative abundances (<0.01 of the total community), none of them was present across samples collected at different times, and they did not include putative bloom-associated taxa *Chaetoceros*, *Gyrodinium*, or *Micromonas*. Thus, bloom-associated marine microalgae do not seem to be plausible producers of the observed INPs.

Further, we used the bacterial community composition to identify potential marine bacterial producers of INPs and to obtain insights into the source environment of the observed INPs. The 16S rRNA analysis showed increasing 16S rRNA gene copy numbers from June to September with a significantly higher alpha diversity of bacteria (observed and Shannon) in June (Fig. S12). A comparably high microbial alpha diversity as we observed in June was previously reported for soils adjacent to Kobbefjord and may point to parts of the community introduced by terrestrial runoff (Šantl-Temkiv et al., 2018). Interestingly, although INPs were concentrated in the SML, bacterial cells did not show a similar pattern, as the differences in copy numbers and alpha diversity between the SML and SBW were not significant. The bacterial community (Fig. S13 and Fig. S14) was dominated by the classes Bacteroidia and Gammaproteobacteria. Throughout all months, we observed a high abundance of ASV affiliated with the genus *Polaribacter*, which was previously found to correlate with the post-bloom and declining stage of the

phytoplanktonic bloom in an Arctic fjord (Feltracco et al., 2021). The principal component analysis (PCA) together with PerMANOVA demonstrated a significant difference between the bacterial community composition in June, July, and September (p < 0.001) (Fig. S15). These findings underscore the seasonal differences in the bacterial community structure with a lower copy number but higher alpha diversity in June when $INP_{-10}$ concentrations were highest.

The results of the CCA for the bacterial community are presented in Fig. 6. The bacterial communities in July and September exhibit a higher degree of similarity between each other than the community observed in June. Additionally, the analysis demonstrated similarities between the communities in the two fjords. The Mantel test was subsequently conducted to assess the correlation between bacterial community dissimilarities (measured using robust Aitchison distance) and environmental parameters. We found that the community composition was strongly correlated with the concentration of $INP_{-10}$ (r = 0.65, p = 0.003). Chlorophyll a was weakly correlated with the bacterial community composition (r = 0.31, p = 0.01), implying that the bloom dynamics had some impact on the community (Table S2). The salinity exhibited a strong correlation (r = 0.67, p = 0.003), emphasizing its influential role in shaping the bacterial community composition (Table S2). Further, we found a strong negative correlation (r = -0.81, p < 0.001) between salinity and the concentration of $INP_{-10}$ with significantly lower salinity but higher concentration of INPs observed in June (Fig. 7a). These correlations suggest a strong impact of salinity within the observed fjords, both impacting the bacterial community composition as well as the $INP_{-10}$ concentrations. The observed freshening of the sea water can be attributed to freshwater input, which may originate either from terrestrial runoff or the melting of sea ice. Terrestrial runoff could either contain INPs produced by terrestrial microorganisms that were introduced into the fjord system from the same source as the bacteria or it might provide nutrients to marine microorganisms, thereby enhancing microbial production of ice nucleation active material in the fjords (Irish et al., 2019; Irish et al., 2017; Meire et al., 2017; Arrigo et al., 2017). Alternatively, sea ice melt water could be a potential source of INPs, however, studies that demonstrate the presence of highly active INPs in sea ice are still lacking.

Using CCA we found a co-occurrence between three taxa, *Aquaspirillum arcticum*, *Colwellia sp.,* and *SUP05* (sulfur-oxidizing Proteobacteria cluster 05) and a high concentration of $INP_{-10}$ in the samples (Fig. 6). As confirmed by Spearman correlation, the co-occurrence was strong between $INP_{-10}$ and the abundance of *Aquaspirillum arcticum* (r = 0.90, p < 0.001) a psychrophilic bacterium found in freshwater Arctic environments (Butler et al., 1989; Brinkmeyer et al., 2004) and *Colwellia* sp. (r = 0.83, p < 0.001) commonly found in sea ice and polar seas (Brinkmeyer et al., 2004). Further, we identified a strong correlation between the presence of ~300 bacterial taxa and the concentration of $INP_{-10}$ (see Dataset (Wieber, 2024)). The most abundant of these bacterial taxa were affiliated to known marine bacterial groups (e.g. SAR11 Clade Ia, Candidatus Aquiluna, Amylibacter...), but were negatively correlated with the $INP_{-10}$, exluding them as potential INP producers. Among the known INA genera, only Pseudomonas was found to correlate with the $INP_{-10}$. The properties of the highly active INPs reported in this study differ from what has been previously reported for ice-nucleating proteins produced by several species of *Pseudomonas* (Hartmann et al.,

2022a; Hara et al., 2016; Garnham et al., 2011), implying that members of this genus were not likely the producers of these INPs. As we found several bacterial taxa correlating with INP$_{-10}$, it is possible that previously unknown INA bacteria, producing INA compounds different from known bacterial ice-nucleating proteins, could be responsible for the ice nucleation activity observed. In the environment, it is likely that the INPs released as exudates (Pummer et al., 2012; Pouleur et al., 1992; Fröhlich-Nowoisky et al., 2015), such as the ones we found predominant during June, may be disassociated from their producer both in the original environments and during their transport to other environments, which may affect the ability to detect both the INP and its producers simultaneously. Therefore, conclusions based on correlations should in cases, where INP exudates are involved, be taken with care and would require further confirmation of putative novel INA microorganisms through cultivation and testing. Alternatively, INPs and bacterial taxa which correlated with the presence of INPs might not be their producers but could have been co-transported to the fjords from terrestrial source environments. This is supported by the significantly positive correlation between INP$_{-10}$ and the presence of bacteria typically associated with soil and terrestrial environments such as *Rhodoferax* (Lee et al., 2022), *Glaciimonas* (Zhang et al., 2011), and *Janthibacterium* (Chernogor et al., 2022) (see Dataset (Wieber, 2024)) as well as the correlation between bacterial diversity and INP$_{-10}$ concentrations (Table S2). Overall, the bacterial community analysis aligns with the conclusion that terrestrial runoff may be the key source of the freshwater input and thus low salinities in June. The timely co-occurrence of high INP concentrations with the post-phytoplanktonic bloom is likely a spurious correlation as terrestrial runoff may also contain nutrients that could stimulate the phytoplanktonic bloom (Juranek, 2022). While the previously presented results indicate that terrestrial runoff is reposible for the reduced salinity observed in June, which correlated to high INP concentrations, we included the analysis of stable oxygen isotopes $\delta^{18}O$ to exclude the possibility of melting sea ice driving the freshening of the seawater.

2.  The authors propose two alternate explanations for the seasonal variations in INP properties proposed by the authors (Lines 382-393). On the other hand, according to explanations related to the analysis of the bacterial community derived from 16S rRNA data (Figs. S6, S7, S12, and S13, and Table S2), it seems that certain mechanisms related to the bacterial community are more important than those related to the eukaryotic community for the seasonal variations in INP properties. Why did the authors exclude the possible contribution of the mechanisms related to the bacterial community from the two explanations?

While INPs produced by bacteria are often membrane bound proteins and thus associated with the cells, the INPs of fungal spores or pollen are often released from the microorganisms and thus not necessarily found in the same environment. The results from the filtration experiments show that the observed INPs are smaller than 0.2µm in all samples and thus likely not associated to bacterial cells. Thus, the community composition is mainly used to get a better understanding of the source environment which leads to the input of the highly active INPs. It would be possible that, so far

unknown INA bacteria, produce INA compounds that different from known bacterial ice-nucleating proteins and have properties similar to the eukaryotic INPs. However, due to the evidence presented in the manuscript it is more likely that the INPs are released by pollen, fungal spores, or lichen in terrestrial environments and transported into the seawater by streams.

We have extended and clarified the analysis of the bacterial community composition as shown in the revised version of section 3.4. Additionally, have made the following changes:

Line 345: "By further characterizing the INPs, we aimed at understanding how their properties fit with properties of INA compounds produced by known INA organisms."

Line 374: "Several studies have shown that while known bacterial ice-nucleating proteins are membrane-bound (Hartmann et al., 2022b; Roeters et al., 2021; Garnham et al., 2011) and thus primarily associated with cells, INPs which were washed off pollen grains and fungal cells are within the size range between 100 and 300 kDa (Pummer et al., 2012; Pouleur et al., 1992; Fröhlich-Nowoisky et al., 2015). Schwidetzky et al. (2023) showed that fungal INPs comprise cell-free proteinaceous aggregates, with 265 kDa aggregates initiating nucleation at -6.8 °C, while smaller aggregates nucleated at lower temperatures. Overall, the properties of the INPs observed in June correspond well with the properties reported for the INA exudates from fungi and pollen and point towards terrestrial environments as a potential source of INPs transported to the seawater."

L407: "The observation that INPs in June are smaller and exhibit distinct responses to heat treatments compared to those observed later in the summer supports the idea that they represent distinct types of INPs. We propose two alternative explanations for the origin of the specific type of INPs observed in June. They could either have been produced by indigenous microbial processes in the seawater or be transported into seawater from terrestrial environments by streams. Based on laboratory studies of known INA organisms, INPs in June may have been released by pollen, fungal spores, or lichen in terrestrial environments and introduced into the seawater by terrestrial runoff. Alternatively, INPs could be yet unknown and uncharacterized molecules produced by marine microorganisms in the late-bloom season. INPs present in July and September have similar properties reported previously by several studies in marine systems indicating that indigenous microbial processes during post-bloom period were responsible for their production. However, INPs observed in July and September could also emerge due to aging processes modifying properties of INPs introduced in June. The increase in INP molecular weight could be due to aggregation in the seawater."

3. I strongly suggest that the authors perform additional analyses and discuss the possible relationship between the variation of the bacterial communities derived from 16S rRNA data and terrestrial runoff. In particular, the authors should compare the bacterial communities found in the sea water samples with those in terrestrial and marine sources, and then evaluate whether the bacterial communities found in the June sea water samples were indeed characterized by terrestrial runoff. In addition, the authors should give more detailed explanations for the reason why the authors focused on the relation between only three taxa (Aquaspirillum arcticum, Colwellia sp., and SUP05) and a high concentration of INPs active at -10 degree C (Lines 431-446) and ignored other taxa.

We have chosen the three taxa (Aquaspirillum arcticum, Colwellia sp., and SUP05) due to the strong correlation with the INP concentration in the CCA analysis, which leads to the conclusion that these bacteria might be transported from the same source environment as the INPs. As discussed in the revised section 3.4 we have extended the analysis and added a list of all taxa that correlated significantly with the INP-10 concentration, which can be accesses by the following link (https://doi.org/10.5281/zenodo.14044414, (Wieber, 2024)). We have manually checked several taxa and those are mainly of terrestrial/freshwater origin. This supports the conclusion that the INPs are indeed transported from terrestrial environments to the sea water.

We kindly refer to the revised version of section 3.4, especially in Lines 481-500.

4. Although the authors explain that "we found a strong negative correlation between salinity and the concentration of INPs active at -10 degree C with significantly lower salinity but higher concentration of INPs observed in June (Fig. 7a) and these correlations suggest a strong impact of salinity within the observed fjords, pointing towards terrestrial runoff or melting sea ice as input of freshwater and potentially INPs (Lines 427-430)", I doubt if there is a possibility that this is a result from the depression of the freezing point caused by salinity. If the authors believe that a negative correlation between salinity and INP abundance suggests freshwater input as sources of INPs (Line 25-26), they should provide evidence that this negative correlation was not caused by the depression of the freezing point.

Thank you very much for this comment, it is indeed important to consider the freezing point depression. However, as described in lines 136-146 ("The salt concentration for each sample was measured using a refractometer (WZ201, Frederiksen scientific, Denmark) and the freezing curves were corrected for the freezing point depression $\Delta T$ using the theoretical formula for sodium chloride.....") we have measured to salt concentrations for all our samples and corrected the values for the effect of the freezing point depression. We can therefore exclude that the negative correlation between the concentration of INPs and the salinity is a result of the freezing point depression.

**References**

Ardyna, M. and Arrigo, K. R.: Phytoplankton dynamics in a changing Arctic Ocean, Nature Climate Change, 10, 892-903, 10.1038/s41558-020-0905-y, 2020.

Arrigo, K. R., van Dijken, G. L., Castelao, R. M., Luo, H., Rennermalm, Å. K., Tedesco, M., Mote, T. L., Oliver, H., and Yager, P. L.: Melting glaciers stimulate large summer phytoplankton blooms in southwest Greenland waters, Geophysical Research Letters, 44, 6278-6285, https://doi.org/10.1002/2017GL073583, 2017.

Balzano, S., Percopo, I., Siano, R., Gourvil, P., Chanoine, M., Marie, D., Vaulot, D., and Sarno, D.: Morphological and genetic diversity of Beaufort Sea diatoms with high contributions from the Chaetoceros neogracilis species complex, J Phycol, 53, 161-187, 10.1111/jpy.12489, 2017.

Biswas, H.: A story of resilience: Arctic diatom Chaetoceros gelidus exhibited high physiological plasticity to changing CO2 and light levels, Frontiers in Plant Science, 13, 10.3389/fpls.2022.1028544, 2022.

Booth, B. C., Larouche, P., Bélanger, S., Klein, B., Amiel, D., and Mei, Z. P.: Dynamics of Chaetoceros socialis blooms in the North Water, Deep Sea Research Part II: Topical Studies in Oceanography, 49, 5003-5025, https://doi.org/10.1016/S0967-0645(02)00175-3, 2002.

Brinkmeyer, R., Glöckner, F.-O., Helmke, E., and Amann, R.: Predominance of β-proteobacteria in summer melt pools on Arctic pack ice, Limnology and Oceanography, 49, 1013-1021, https://doi.org/10.4319/lo.2004.49.4.1013, 2004.

Butler, B. J., McCallum, K. L., and Inniss, W. E.: Characterization of Aquaspirillum arcticum sp. nov., a New Psychrophilic Bacterium, Systematic and Applied Microbiology, 12, 263-266, https://doi.org/10.1016/S0723-2020(89)80072-4, 1989.

Chernogor, L., Bakhvalova, K., Belikova, A., and Belikov, S.: Isolation and Properties of the Bacterial Strain Janthinobacterium sp. SLB01, Microorganisms, 10, 10.3390/microorganisms10051071, 2022.

Feltracco, M., Barbaro, E., Hoppe, C. J. M., Wolf, K. K. E., Spolaor, A., Layton, R., Keuschnig, C., Barbante, C., Gambaro, A., and Larose, C.: Airborne bacteria and particulate chemistry capture Phytoplankton bloom dynamics in an Arctic fjord, Atmospheric Environment, 256, 118458, https://doi.org/10.1016/j.atmosenv.2021.118458, 2021.

Fröhlich-Nowoisky, J., Hill, T. C. J., Pummer, B. G., Yordanova, P., Franc, G. D., and Pöschl, U.: Ice nucleation activity in the widespread soil fungus Mortierella alpina, Biogeosciences, 12, 1057-1071, 10.5194/bg-12-1057-2015, 2015.

Garnham, C. P., Campbell, R. L., Walker, V. K., and Davies, P. L.: Novel dimeric β-helical model of an ice nucleation protein with bridged active sites, BMC Structural Biology, 11, 36, 10.1186/1472-6807-11-36, 2011.

Hara, K., Maki, T., Kakikawa, M., Kobayashi, F., and Matsuki, A.: Effects of different temperature treatments on biological ice nuclei in snow samples, Atmospheric Environment, 140, 415-419, https://doi.org/10.1016/j.atmosenv.2016.06.011, 2016.

Hartmann, S., Ling, M., Dreyer, L. S. A., Zipori, A., Finster, K., Grawe, S., Jensen, L. Z., Borck, S., Reicher, N., Drace, T., Niedermeier, D., Jones, N. C., Hoffmann, S. V., Wex, H., Rudich, Y., Boesen, T., and Santl-Temkiv, T.: Structure and Protein-Protein Interactions of Ice Nucleation Proteins Drive Their Activity, Front Microbiol, 13, 872306, 10.3389/fmicb.2022.872306, 2022a.

Hartmann, S., Ling, M., Dreyer, L. S. A., Zipori, A., Finster, K., Grawe, S., Jensen, L. Z., Borck, S., Reicher, N., Drace, T., Niedermeier, D., Jones, N. C., Hoffmann, S. V., Wex, H., Rudich, Y., Boesen, T., and Santl-

Temkiv, T.: Structure and Protein-Protein Interactions of Ice Nucleation Proteins Drive Their Activity, Front Microbiol, 13, 10.3389/fmicb.2022.872306, 2022b.

Hegseth, E. N. and Sundfjord, A.: Intrusion and blooming of Atlantic phytoplankton species in the high Arctic, Journal of Marine Systems, 74, 108-119, https://doi.org/10.1016/j.jmarsys.2007.11.011, 2008.

Irish, V. E., Hanna, S. J., Xi, Y., Boyer, M., Polishchuk, E., Ahmed, M., Chen, J., Abbatt, J. P. D., Gosselin, M., Chang, R., Miller, L. A., and Bertram, A. K.: Revisiting properties and concentrations of ice-nucleating particles in the sea surface microlayer and bulk seawater in the Canadian Arctic during summer, Atmos. Chem. Phys., 19, 7775-7787, 10.5194/acp-19-7775-2019, 2019.

Irish, V. E., Elizondo, P., Chen, J., Chou, C., Charette, J., Lizotte, M., Ladino, L. A., Wilson, T. W., Gosselin, M., Murray, B. J., Polishchuk, E., Abbatt, J. P. D., Miller, L. A., and Bertram, A. K.: Ice-nucleating particles in Canadian Arctic sea-surface microlayer and bulk seawater, Atmos. Chem. Phys., 17, 10583-10595, 10.5194/acp-17-10583-2017, 2017.

Johnsen, G. and Sakshaug, E.: Bio-optical characteristics and photoadaptive responses in the toxic and bloom-forming dinoflagellates Gyrodinium aureolum, Gymnodinium galatheanum, and two strains of Prorocentrum minimum, Journal of Phycology, 29, 627-642, https://doi.org/10.1111/j.0022-3646.1993.00627.x, 1993.

Juranek, L. W.: Changing biogeochemistry of the Arctic Ocean: Surface nutrient and CO2 cycling in a warming, melting north, Oceanography, 35, 144–155, https://doi.org/10.5670/oceanog.2022.120, 2022.

Juul-Pedersen, T., Arendt, K. E., Mortensen, J., Blicher, M. E., Søgaard, D. H., and Rysgaard, S.: Seasonal and interannual phytoplankton production in a sub-Arctic tidewater outlet glacier fjord, SW Greenland, Marine Ecology Progress Series, 524, 27-38, 2015.

Lee, Y. M., Park, Y., Kim, H., and Shin, S. C.: Complete genome sequence of Rhodoferax sp. PAMC 29310 from a marine sediment of the East Siberian Sea, Marine Genomics, 62, 100891, https://doi.org/10.1016/j.margen.2021.100891, 2022.

Marquardt, M., Vader, A., Stübner, E. I., Reigstad, M., and Gabrielsen, T. M.: Strong Seasonality of Marine Microbial Eukaryotes in a High-Arctic Fjord (Isfjorden, in West Spitsbergen, Norway), Appl Environ Microbiol, 82, 1868-1880, doi:10.1128/AEM.03208-15, 2016.

Meire, L., Mortensen, J., Meire, P., Juul-Pedersen, T., Sejr, M. K., Rysgaard, S., Nygaard, R., Huybrechts, P., and Meysman, F. J. R.: Marine-terminating glaciers sustain high productivity in Greenland fjords, Global Change Biology, 23, 5344-5357, https://doi.org/10.1111/gcb.13801, 2017.

Pouleur, S., Richard, C., Martin, J. G., and Antoun, H.: Ice Nucleation Activity in Fusarium acuminatum and Fusarium avenaceum, Appl Environ Microbiol, 58, 2960-2964, 10.1128/aem.58.9.2960-2964.1992, 1992.

Pummer, B. G., Bauer, H., Bernardi, J., Bleicher, S., and Grothe, H.: Suspendable macromolecules are responsible for ice nucleation activity of birch and conifer pollen, Atmos. Chem. Phys., 12, 2541-2550, 10.5194/acp-12-2541-2012, 2012.

Roeters, S. J., Golbek, T. W., Bregnhoj, M., Drace, T., Alamdari, S., Roseboom, W., Kramer, G., Santl-Temkiv, T., Finster, K., Pfaendtner, J., Woutersen, S., Boesen, T., and Weidner, T.: Ice-nucleating proteins are activated by low temperatures to control the structure of interfacial water, Nat Commun, 12, 1183, 10.1038/s41467-021-21349-3, 2021.

Šantl-Temkiv, T., Gosewinkel, U., Starnawski, P., Lever, M., and Finster, K.: Aeolian dispersal of bacteria in southwest Greenland: their sources, abundance, diversity and physiological states, FEMS Microbiol Ecol, 94, 10.1093/femsec/fiy031, 2018.

Schwidetzky, R., de Almeida Ribeiro, I., Bothen, N., Backes, A. T., DeVries, A. L., Bonn, M., Fröhlich-Nowoisky, J., Molinero, V., and Meister, K.: Functional aggregation of cell-free proteins enables fungal ice nucleation, Proceedings of the National Academy of Sciences, 120, e2303243120, doi:10.1073/pnas.2303243120, 2023.

Terhaar, J., Lauerwald, R., Regnier, P., Gruber, N., and Bopp, L.: Around one third of current Arctic Ocean primary production sustained by rivers and coastal erosion, Nature Communications, 12, 169, 10.1038/s41467-020-20470-z, 2021.

Vader, A., Marquardt, M., Meshram, A. R., and Gabrielsen, T. M.: Key Arctic phototrophs are widespread in the polar night, Polar Biology, 38, 13-21, 10.1007/s00300-014-1570-2, 2015.

Wieber, C., Jensen, L. Z., Vergeynst, L., Maire, L., Juul-Pedersen, T., Finster, K., & Šantl-Temkiv, T.: Dataset to: Terrestrial runoff is the dominant source of a new type of biological INPs observed in Arctic fjords [Data set], Zenodo, https://doi.org/10.5281/zenodo.14044414, 2024.

Zhang, D.-C., Redzic, M., Schinner, F., and Margesin, R.: Glaciimonas immobilis gen. nov., sp. nov., a member of the family Oxalobacteraceae isolated from alpine glacier cryoconite, International Journal of Systematic and Evolutionary Microbiology, 61, 2186-2190, https://doi.org/10.1099/ijs.0.028001-0, 2011.